

# A semantic enhanced clothing recommendation system based on implicit trust graphs and ontology web language

Luyao He[1], Huazhi Xiang[2], Pietro Alex Marra[3] and Chunjia Wang[4]

[1] Hubei Polytechnic University, Huangshi, Hubei, China
[2] Hubei Business College, Wuhan, Hubei, China
[3] Academy of Fine Arts of Florence, Florence, Tuscany, Italy
[4] West Yunnan University of Applied Sciences, Tengchong, Yunnan, China

Corresponding author
Luyao He, heluyaoedu@163.com

## ABSTRACT

In recent years, online clothing shopping has gained significant popularity due to its convenience and the vast range of products available to consumers. However, finding apparel that aligns with individual preferences remains a challenge for many shoppers, leading to time-consuming searches and unsatisfactory experiences. Traditional recommendation systems, such as collaborative filtering (CF), offer personalized clothing suggestions by analyzing user behavior and preferences. Despite their widespread use, these systems suffer from inherent limitations, including data sparsity, cold start issues, and an inability to accommodate users with multiple interests across diverse clothing categories effectively. To address these challenges, this article proposes a semantic-enhanced trust clothing recommendation system based on ant colony optimization, which overcomes the limitations of traditional recommendation systems when considering user context and trust. This system fully utilizes the collective intelligence of ant colony optimization algorithms, combined with the semantic information of clothing products and user trust models, to provide users with more personalized and reliable clothing recommendations. Specifically, the clothing products are initially modeled through semantic description, converting clothing features into semantic vectors to create a semantic similarity model. At the same time, a trust model is introduced to consider user historical behavior and semantic similarity, and to calculate the user's trust in other users. Subsequently, the system simulates users with individual ants and uses ant colony optimization algorithms to simulate the collaborative behavior of ants in information search and transmission. The experimental results show that the system has significant advantages over traditional methods in providing personalized and reliable clothing recommendations. The system's real-time updates and learning mechanism ensure continuous optimization of recommendation performance, making it better adapted to dynamic changes in user clothing interests.

# INTRODUCTION

Online shoppers are increasingly opting to purchase apparel due to the convenience offered by companies providing a wide range of products (*Nagar & Gandotra, 2016*). Data from 2020 indicates that approximately 80% of all transactions in China's online clothing businesses occurred online. However, when it comes to online buying, consumers often encounter a common problem: they find it challenging to locate clothing products that precisely match their preferences, leading to a significant time investment in exploration (*Liu et al., 2018*). Therefore, to enhance the purchasing experience for customers, online clothing merchants must provide personalized apparel recommendations. Users and clothing products are the two main components of any clothing recommendation system. Active users are those who utilize the recommendation system and rate various clothing products to express their opinions. The recommendation system processes the rating data input using filtering algorithms to offer suggestions for new clothing products (*i.e.*, target items). This enables the system to furnish active users with personalized clothing recommendations.

Collaborative filtering (CF) recommendation algorithms employ groups with similar clothing interests to provide clothing recommendations to target individuals (*Shokrzadeh et al., 2024*; *Pugoy & Kao, 2024*). However, they also have to deal with issues like data sparsity and the cold start problem (*Hassan, Rafi & Frnda, 2024*; *Kumari & Sagar, 2023*; *Li et al., 2021*; *Natarajan et al., 2020*). Data sparsity occurs when there are significantly fewer user ratings available than the ratings that need to be anticipated. This makes it challenging to find a substantial overlap between clothing products evaluated by two consumers (*Berkani, 2020*).

In the domain of apparel recommendation systems, the user-based collaborative filtering (UCF) algorithm represents a conventional recommendation methodology (*Chen et al., 2020*; *Jain et al., 2020*). The fundamental principle involves generating Top-K neighbors based on the user's historical apparel selections. Subsequently, personalized clothing recommendations are offered to the target user in accordance with the preferences of these Top-K neighbors (*Liu & Shi, 2023*). Nonetheless, the UCF algorithm exhibits certain limitations, such as sparse matrices, diminished efficiency, reduced accuracy, and limited recommendation coverage. To tackle these challenges, an advanced user-based collaborative filtering (AUCF) algorithm is introduced, with the primary objective of mitigating matrix sparsity while enhancing recommendation coverage and efficiency (*Wang et al., 2021*; *Martins, Papa & Adeli, 2020*). Through rigorous 10-fold cross-validation experiments conducted on the Tmall dataset, the findings indicate that the AUCF algorithm has manifested noteworthy advancements in augmenting recommendation coverage and accuracy. Within the realm of clothing recommendation systems, the AUCF algorithm emerges as a viable solution for elevating the comprehensiveness and precision of recommendations.

In addition to the issues mentioned earlier, another limitation of collaborative filtering (CF) is its inability to cater to users involved in multiple interests and diverse clothing products. In the context of clothing recommendation, this means that traditional CF methods may not effectively consider the diverse preferences of users in various clothing

categories or styles, thus affecting the accuracy and personalization of recommendations. In such cases, the recommendation effectiveness of collaborative filtering is compromised because the target clothing preferences of active users may differ from the shared interests of their neighbors. This issue is referred to as multiple interests and multiple content (MIMC) (*Li & Han, 2020*). Acknowledging the inherent challenges associated with collaborative filtering (CF) techniques, numerous researchers have redirected their focus towards hybrid systems incorporating supplementary external data, such as trust information (*Varlamis et al., 2022*), semantic details, and demographic insights. Semantically driven CF systems have recently demonstrated notable success across diverse domains. These systems exhibit the capacity to offer more nuanced recommendations, particularly in scenarios involving highly sparse datasets or novel initiatives, by leveraging semantic reasoning. In recent years, semantic-oriented recommendation systems have found application in diverse industries, encompassing health, travel/leisure, news, sound/movie/music, and beyond (*Li et al., 2023*; *De Gemmis et al., 2015*; *Berkani et al., 2021*). Furthermore, as an innovative avenue of research, multiple studies propose that addressing data sparsity and mitigating cold start challenges can be achieved by infusing social trust information into conventional collaborative filtering techniques, resulting in higher-quality recommendations. The amalgamation of semantic data with social trust data is expected to enhance the performance of recommendation systems in clothing suggestions, especially when dealing with diverse clothing categories and user interest complexities.

To address the issues above, this article proposes an innovative ant recommendation system based on a semantic-enhanced trust. This system combines semantic information and trust mechanisms to improve the performance of recommendation systems. Through the collective intelligence of ant colony optimization, the system generates a semantically enhanced clothing recommendation list and dynamically recommends based on the weight of the user's historical behavior, considering time information. The experimental results indicate that the system has significant advantages over traditional methods in providing personalized and reliable clothing recommendations. The system's real-time updates and learning mechanism ensure the continual optimization of recommendation performance, enhancing its adaptability to dynamic shifts in user clothing interests. The primary contributions of this article encompass:

(1) Semantic-enriched recommendation: Integration of semantic information to empower recommendation systems in comprehending the intricate relationships between users and items. The utilization of semantic information enables the system to more precisely grasp user interests and item characteristics, consequently delivering more accurate recommendations.

(2) Trust mechanism application: Incorporation of a trust mechanism to address the challenges of data sparsity and cold start encountered in traditional collaborative filtering methods. By factoring in the trust relationships among users, the system optimally utilizes existing information, thereby elevating the accuracy and personalization of recommendations.

(3) Ant algorithm optimization: Innovative utilization of the ant algorithm to refine the recommendation process, imparting greater flexibility to the recommendation system in adapting to diverse environments and user requirements. The introduction of the ant algorithm enhances system robustness and efficiency.

(4) Through the amalgamation of semantic enhancement and trust mechanisms, the system proposed in this article presents a holistic solution adept at overcoming multiple challenges inherent in traditional recommendation systems, including data sparsity, diverse interests, and varied content.

## RELATED WORKS

The relevant topics of the recommended recommendation system will be discussed in this section. First, the ant colony algorithm was briefly introduced. After that, we reviewed the relevant research on hybrid and CF recommendation systems.

### Ant colony optimization

Group intelligence represents a behavioral and computational paradigm for problem-solving, drawing inspiration from the social behavior observed in insects and other animals. The ant colony algorithm stands out as a premier optimization strategy that closely mimics the real-world foraging behavior of ants (*Majid, Arshad & Mokhtar, 2022*). Ants use covert communication to determine the shortest path, reinforcing it with increased pheromones because it requires less time to traverse. Consequently, these shorter paths accrue preference over time. It is important to note that pheromones dissipate over time. Therefore, longer paths gradually lose their pheromone potency and attractiveness. Eventually, the majority of ants will identify the fastest route from their colony to their food supply. Various Ant Colony Optimization (ACO) algorithms adopt a similar approach in tackling optimization problems (*Bhavya & Elango, 2023*; *Revanna & Al-Nakash, 2023*). Throughout the iteration process, the optimal solution to a problem is identified. The solutions discovered by the ants in each iteration serve as a blueprint for generating new solutions in subsequent iterations. To elaborate, each iteration fine-tunes the pheromone trail, directing ants toward paths more inclined to yield feasible solutions. This iterative process persists until the predefined goal is achieved.

### CF recommender systems

One of the most commonly utilized technologies in recommender systems is the collaborative filtering (CF) method. CF can be broadly classified into memory-based techniques and model-based methods. Model-based approaches involve creating a model based on past ratings, whereas memory-based techniques utilize the entire rating matrix to offer recommendations (*Isaoglu & Yiltas-Kaplan, 2023*). Within memory-based techniques, user-based CF (UCF) (*Ma et al., 2023*) and item-based CF (ICF) (*Abdalla et al., 2023*) represent two additional categories. In the UCF approach, neighborhoods, which are subsets of individuals, are selected by comparing users to active users. Subsequently, the ratings of active users are forecasted by employing a weighted combination of neighbor ratings. The ICF approach mirrors the UCF approach, albeit relying on the similarity

between items rather than users (*Tanwar & Vishwakarma, 2023*). A hybrid technique, proposed in *Li, Lu & Xuefeng (2005)*, combines UCF and ICF methods to tackle issues related to data sparsity and cold start in CF methods. This technique selects neighbors of active users based on items similar to the target and filters out dissimilar items. *Choi & Suh (2013)* introduces a novel similarity function that identifies superior neighbors based on each unique target item. The suggested function assigns weights to the user's assessment of an item according to its similarity to the target item. Over the past few decades, researchers have developed various techniques to address the challenges of cold start and data sparsity. For instance, *Ahmed & Letta (2023)* systematically compared memory-based and model-based collaborative filtering algorithms in the scenario of book recommendation. Their results indicated that under data-sparse conditions, such as cold start, model-based methods like matrix factorization (SVD) demonstrate stronger robustness than k-nearest neighbors (KNN)-based methods.

## Semantic-based CF recommender systems

In recent years, introducing semantics into CF recommendation systems has been a viable approach to addressing issues such as data sparsity (*Alhijawi et al., 2022*). Traditional semantic networks lose much helpful information in recommendation systems. As such, only things that are strikingly comparable to what the user already knows are included in their suggestions. By determining implicit semantic links between things, semantic-based recommendation algorithms can get around this issue. The semantic web relies on taxonomies or ontologies to categorize and describe concepts within a specific domain. The system utilizes product classification and ontology to derive the semantics of products, revealing potential semantic connections between them. Compared to traditional CF methods, semantic-based CF methods have two key advantages. Firstly, the system can infer potential reasons why users are interested or not interested in a specific project based on its semantic attributes. Secondly, by fully utilizing semantic information, the system can still provide meaningful recommendations to users even in the presence of new projects or highly sparse datasets (*Alhijawi et al., 2022*).

## Trust-based CF recommender systems

Trust information is frequently integrated into the collaborative filtering (CF) method as an additional strategy to address the challenges of data sparsity and cold start in traditional CF methodologies. Typically, resulting hybrid systems explore trust networks, considering users directly or indirectly trusted by a given user. These systems aggregate evaluations from those neighbors to generate recommendations. While extending trust to indirect neighbors in social networks, most trust-based recommendation systems leverage the transitivity of trust. However, many proposed approaches often overlook the contextual interdependence and dynamic nature of trust. This discussion delves into several prominent implicit trust-based filtering techniques (*Jing et al., 2024*).

In *Pitsilis & Marshall (2004)*, a method is introduced to implicitly infer user trust from data detailing how they rate items. This paradigm expresses trust through opinions, which can occasionally be ambiguous and subjective. Formulating a model for calculating trust in

*Inferences (2005)* considers the subjective nature of trust relationships by incorporating confidence and uncertainty attributes. *Huang et al. (2025)* introduces a robust k-nearest recommendation method, allowing users to evaluate the reliability of the rating information they receive, thereby determining the level of mutual trust. The trust-based similarity fusion (TSF) recommendation approach outperforms user- and item-based benchmark algorithms in terms of prediction accuracy and coverage (*Hwang & Chen, 2007*). However, it does not consider contextual information in its experiments, and the model overlooks the dynamic aspects of trust.

Recent research suggests that trust-based recommendation systems employing ant colony algorithms may address the dynamics of trust. To address this issue, dynamic trust-based ant recommender (DT-BAR) (*Bellaachia & Alathel, 2014*) is proposed as a dynamic trust-based recommender, allowing ants to exchange traversal edge information to tackle new user challenges. Another shortcoming in T-BAR is the absence of consideration for contextual information. This research introduces a novel implicit trust-based filtering technique called Ant Trust-based Semantic-enhanced Recommendation System (ATSRS) to address these issues. ATSRS calculates trust by incorporating contextual information and possesses attributes of asymmetry, transmission, and dynamics. The algorithm employs the ant colony optimization approach to search for the optimal trust path in the trust network in a depth-first manner, enabling it to determine the best neighbor for active users.

## METHODOLOGY

This section presents a comprehensive introduction to the Ant Trust-based Semantic-enhanced Recommendation System (ATSRS). ATSRS stands out as a dynamic recommendation system that factors in user context, specifically focusing on user personality preferences and time information. It strategically selects the most trustworthy neighbor based on the current interests of active users in a particular type of clothing. To infer contextually relevant trust values, the system leverages the semantic descriptions associated with clothing products. In practical implementation, ATSRS organizes projects into clusters based on semantic similarity and identifies trusted neighbors of active users within a specific cluster.

We define the relevant symbols as follows: represents a given set of items. Represents a given set of users. By clustering clothing products based on their semantic similarity, we obtained a set of z clothing product clusters denoted as. Throughout the remainder of this article, whenever the context is mentioned, it refers to the corresponding cluster.

### The structure of ATSRS

As shown in Fig. 1, ATSRS consists of a database and a recommendation engine. A database is a system or collection that organizes and stores data. In this system, a database contains user information (such as personal information), characteristics and descriptions of clothing products, and historical ratings of users. A recommendation engine is a software system that generates personalized recommendations based on user preferences, historical behavior, or other characteristics. Recommendation engines may use user and clothing information from databases to analyze and predict clothing products that users

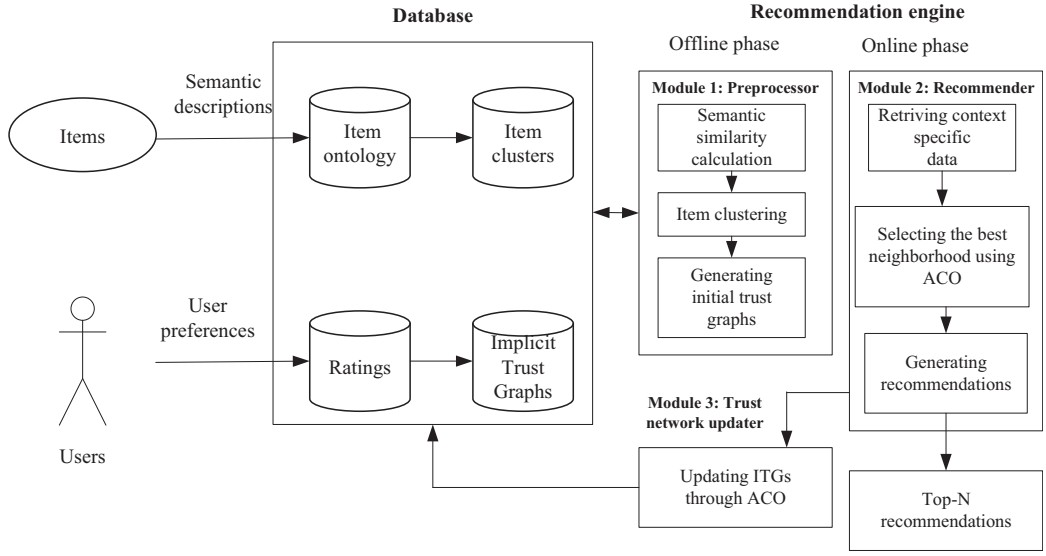

**Figure 1 The architecture of the ATSRS approach.**

may like through different algorithms. In this article, the recommendation engine consists of an offline stage and an online stage. In the offline phase, using a preprocessor module, semantic similarity calculation is first performed based on the clothing information in the dataset, item clustering is carried out, and finally, an initial trust map is generated. In the online stage, a recommendation system was designed. First, retrieve context-specific data. Then, use an ant colony algorithm to select the best neighborhood, update the trust network program, and finally, generate suggestions. Display the top N suggestions based on their priority, allowing users to choose from them. Next, we will provide a detailed introduction to the various parts and processes of the algorithm.

## Preprocessor module

This module utilizes semantic profiles of garment projects to incorporate contextual information into the ATSRS system. To achieve this, clothing products are grouped into clusters based on semantic similarity, and implicit trust graphs (ITGs) are constructed for each cluster. The main concept of this module is that the content of the current clothing products should influence the degree of trust between two users. By computing implicit trust levels derived from the semantic aspects of clothing products, ATSRS systems demonstrate adaptability to a wide range of user interest scenarios, as shown in Fig. 1. Indeed, there is a positive correlation between implicit trust and user interests. Consequently, there may be several trust connections between users when they are engaged in other spheres of interest and clothing products with very distinct content.

Semantic similarity calculation: The aspects of clothing products must be initially represented using a domain ontology to leverage the semantic information contained within them. We implement the ontology of clothing products, encompassing common concepts and relationships, in our clothing recommendation system. Although our approach could apply to other domains with various project ontologies, we chose to

employ the clothes domain ontology in clothing recommendation systems. In the context of apparel, ontology might encompass ideas such as brand, type, color, material, and their connections.

Using domain-specific data, our clothes ontology is constructed using the Ontology Web Language (OWL) standard. This ontology uses standardized language to provide a coherent description of clothing-related ideas and occurrences. Through the ontology, a hierarchical structure can be established among concepts, such as subclasses of clothing types, properties of colors, and materials. Based on their semantic descriptions, two clothing articles, a and b, can have their similarity computed. The following formula can be used to determine semantic similarity.

$$S(a, b) = \sum_{i=1}^{|Q|} \left( \frac{\text{common}(a, b, Q[i])}{\max(\deg(a, Q[i]), \deg(b, Q[i]))} \right) \times Weight(Q[i]) \tag{1}$$

where q is a vector that represents the "target" of recommendation engines and consists of data type attributes and clothing class object properties. Deg(a, b) is a measure of how many instances of item an are connected to attribute q. The number of common occurrences connected to items a and b by attribute q is indicated by the expression common (a, b, q). The Weight(q) represents the importance of attribute q. The assigned domain should be used to decide the weights of the qualities subjectively.

Item clustering: We implemented the k-medoids technique for semantic clustering of projects due to its high accuracy and simplicity. K-medoids is a partition-based clustering approach that is comparable to the k-means algorithm. In contrast to k-means, k-medoids can handle the distance matrix between any items and choose one object as the center rather than averaging the objects. As a result, it is more resilient to anomalies and noise. Additionally, the sequence in which items appear usually has little bearing on k-medoids. In this article, we used the k-medoids technique to preserve the semantic content of the objects while clustering. Firstly, as the initial medium, k terms are randomly selected *via* the k-medoids method. The two stages are then alternated. Assign each item to the cluster linked to the closest medium in the first phase. More specifically, the proximity between two things is measured using semantic similarity as a distance metric. The formula to get the project distance is used to compute the distance between two projects, a and b. Next, replace every non-medoid item in each cluster with a medoid. The item is chosen as the new media item if it is a non-media item used as media and the overall distance within the cluster decreases. The algorithm repeats these stages until the medium is fixed.

Generating initial trust graphs: The preprocessing module initiates the generation of a directed implicit trust graph for each experimental cluster after the project cluster is created. The specific steps are outlined as follows:

Assumption 1. Step 1: Data normalization: Normalizing ratings from various users to the same scale is vital since ratings rely not only on the interests of the user but also on their rating patterns. In this process, we'll normalize the data using the maximum-minimum approach.

$$r' = \frac{r - \min(X)}{\max(X) - \min(X)} \tag{2}$$

where the least and maximum ratings in dataset X are denoted, respectively, by min(X) and max(X).

Step 2: Calculate initial trust level: The local trust model predicts customized trust scores based on the individual and subjective perspectives of each user. In this phase, the clothing recommendation system calculates the direct implicit trust score for each user pair using a rating matrix. More specifically, by averaging the prediction errors of jointly evaluated clothing products, the system determines local trust scores. Two users do not have a direct trust relationship if they have not assessed the same clothes project.

In this article, we first define $I_u^{c_j}(t_0)$ as the rating set of the user $c_j$ prior to a given time point, and $\left|I_u^{c_j}(t_0)\right|$ denotes the number of items rated by the user $c_j$ within this set.

Furthermore, considering the normalized rating results from step 1 and the direct trust score assigned by user u to user v within the cluster context, the calculation is as follows:

$$trust_{u \rightarrow v}^{c_j}(t_0) = \left(1 - \frac{\sum_{i \in D \cap E} (p_{u,i}^v - r_{u,i})^2}{|D \cap E|}\right) \frac{|D \cap E|}{|D| + |E| - |D \cap E|} \tag{3}$$

where $0 < trust_{u \rightarrow v}^{c_j}(t_0) < 1$, $D = I_u^{c_j}(t_0)$, $E = I_v^{c_j}(t_0)$, $D \cap E$ is the set of items evaluated jointly by users u and v in clustering $c_j$, where $p_{u,i}^v$ is the anticipated rating of item i for user u based purely on user v's neighborhood.

$$p_{u,i}^v = \overline{r_u} + \left(r_{v,i} - \overline{r_v}\right) \tag{4}$$

where the mean ratings of users u and v for products in cluster $c_j$ are denoted by $\overline{r_u}$ and $\overline{r_v}$, respectively.

Step 3: Determine the initial global trust: In this phase, ATSRS calculates each user v's global trust score by averaging the local trust scores provided by their immediate neighbors in the trust network. The calculation is as follows:

$$Gtrust_v^{c_j}(t_0) = \frac{1}{\left|NB_v^{c_j}\right|} \sum_{u \in NB_v^{c_j}} trust_{u \rightarrow v}^{c_j}(t_0) \tag{5}$$

where $0 < Gtrust_v^{c_j}(t_0) < 1$.

Step 4: Establish the first directed ITG. The first stage is to construct an initial structure, where the user's collection of vertices is V, and the set of edges linking them is E. The article on the worldwide score follows this.

$$\forall u \in U, \quad u \neq v \quad W_{uv}^{c_j}(t_0) = Gtrust_v^{c_j}(t_0) \tag{6}$$

where $W_{uv}^{c_j}(t_0)$ is the weight information in the link. Therefore, over time, we can collect data on the preferences of more users towards the items in the set $c_j$. Then adjust the weight information online.

To address the critical issue of data sparsity and cold-start scenarios where users have no co-rated items, we introduce an indirect trust propagation mechanism. This mechanism prevents users from remaining as isolated nodes in the trust graph by inferring Trust through semantic and network pathways. Specifically, if a direct trust relationship does not exist between users u$u$ and v$v$ ($I_{u,v}^c = 0$), the system computes an indirect trust

score. This is achieved by identifying all paths of length $2k \leq 2$ connecting $uu$ to $vv$ in the global trust graph. The trust along each path is calculated as the product of the direct trust scores of its constituent edges, with each score weighted by the semantic similarity of the cluster contexts in which those trusts were established. The final indirect trust score for $u \rightarrow v$ is the maximum (or average) of the trust values over all such paths. This approach leverages the transitivity of trust while tempering it with contextual relevance. This ensures that inferred relationships are meaningful, thereby improving the recommendation system's ability to handle new users and sparse data conditions.

## Recommender module

This module is tasked with retrieving context-specific data, assessing it, and employing metaphors from ant colonies to choose the optimal community. Subsequently, it suggests complementary apparel products that customers might find appealing. A detailed explanation of this module's procedure will be provided in the following sections to enhance our understanding of clothing recommendation systems.

Retrieve data based on context: We need to create a recommendation table for active users during the online phase. Firstly, retrieve the relevant rating and trust information from the database. The contextual information of user u at time interval t is as follows, assuming. $\tilde{I}_u(t) = \{i | i \in I \text{ and } r_{u,i} = null\}$ is the set of items the user acquired before time t. The contextual data regarding user u at time interval t is $T_u(t) = \{\text{ITG}^c(t) | c \in \tilde{C}_u(t)\}$, where each graph belongs to a target cluster $c_j$, represents a collection of trust graphs needed for active user u to create recommendations at time interval t.

The set of rating submatrices $R_u(t) = \{\boldsymbol{R^c}(t) | c \in \tilde{C}_u(t)\}$ is used to generate recommendations for active users u with a time interval t. The normalized user ratings that are available for the deadline t in the target cluster c are contained in the sub-matrix $\boldsymbol{R^c}(t)$.

The collection of timestamp submatrices $S_u(t) = \{\boldsymbol{S^c}(t) | c \in \tilde{C}_u(t)\}$ is used to create recommendations for user u at time t. The corresponding time intervals are rated in $\boldsymbol{R^c}(t)$ are contained in the submatrix $\boldsymbol{S^c}(t)$.

## Using the ant colony algorithm to select the best neighborhood

In this study, Ant Colony Optimization (ACO) was adopted over other bio-inspired algorithms for several reasons. ACO is particularly suited for dynamic and decentralized decision-making, which mirrors the collaborative yet individualized nature of human clothing preferences. Unlike algorithms such as Particle Swarm Optimization (PSO) or Genetic Algorithms (GA), ACO inherently models the exploration-exploitation trade-off through pheromone updates, enabling rapid adaptation to evolving user behaviors. Furthermore, empirical studies have demonstrated that ACO outperforms other swarm intelligence models in tasks requiring personalized search and dynamic trust evaluation (*Bhavya & Elango, 2023*). Thus, ACO provides a biologically plausible and computationally efficient framework to optimize trust-based recommendation in highly dynamic contexts such as clothing preferences.

Ant Colony Algorithm Process: The steps of the ant colony algorithm are as follows. In the initial scenario, it can be assumed that all ants are distributed in various positions and the pheromone concentration on each connecting path is the same, set as $\tau_{ab}^c(t)$, *i.e.*,

$$\tau_{ab}^c(t) = W_{ab}^c(t). \tag{7}$$

We set the probability of the k-th ant in the population transferring from user a to user b at time t to $P_{a,b}^k(t)$. Additionally, we specify that ants can only access users they do not know, so we use $A_k$ to represent the set of users that the k-th ant is allowed to access. In the ant colony algorithm, we define $P_{a,b}^k(t)$ as follows:

$$P_{a,b}^k(t) = \begin{cases} \dfrac{[\tau_{a,b}(t)]^\alpha \cdot [\eta_{a,b}(t)]^\beta}{\sum_{s \in A_k} [\tau_{a,s}(t)]^\alpha \cdot [\eta_{a,s}(t)]^\beta}, & if \ b \in A_k \\ 0, & otherwise \end{cases} \tag{8}$$

where $\eta_{a,b}(t)$ represents the reciprocal of the distance between user a and user b, $\alpha$ and $\beta$ are used to represent the importance level between the concentration of pheromones and the heuristic value, respectively.

Pheromone update: In the g-th iteration, the update formula for pheromones is as follows:

$$\tau_{a,b}(g+1) = (1 - \rho) \cdot \tau_{a,b}(g) + \sum_{t=1}^{n} \Delta\tau_{a,b}(t) \tag{9}$$

where $(1 - \rho) \cdot \tau_{a,b}(g)$ describes the decay of pheromones along the path over time t, $0 < \rho < 1$. And $\Delta\tau_{a,b}(t)$ describes that at time t, the pheromones on the path from user a to user b increase due to the passage of ants:

$$\Delta\tau_{a,b}(t) = \sum_{k=1}^{m} \Delta\tau_{a,b}^k(t) \tag{10}$$

where $\Delta\tau_{a,b}^k(t)$ represents the pheromone left by the k-th ant on the path from user a to user b at time t. If the k-th ant does not pass through this path at time t, the increase in pheromone caused by it is recorded as 0. Otherwise, the increase in pheromone caused by it is inversely proportional to the path it has already passed through. We describe this logic using the following formula:

$$\Delta\tau_{a,b}^k(t) = \begin{cases} \dfrac{Q}{L_k}, & if \ ant \ k \ travels \ from \ a \ to \ b \ in \ time \ t \\ 0, & otherwise \end{cases} \tag{11}$$

where $L_k$ does the k-th ant take the total path length during this iteration, and Q is the pheromone constant. Another way to write the above equation is

$$\Delta\tau_{a,b}^k(t) = \begin{cases} \dfrac{Q}{\sum_{s=0}^{m} d_k(s)}, & if \ ant \ k \ travels \ from \ a \ to \ b \ in \ time \ t \\ 0, & otherwise \end{cases}. \tag{12}$$

Updates to the pheromone values based on the chosen neighbors are subsequently necessary. More precisely, at each time step, the pheromone update process follows the neighborhood creation phase. Through this procedure, the pheromone value of the best neighbor for an active user in a specific context increases at each time step. In contrast, the pheromone value of other users decreases.

To formally justify the application of ACO for neighbor selection and resolve the conceptual analogy between path length and user proximity, we establish a precise mathematical mapping. In our model, the fundamental ACO concept of "minimizing path length" is redefined to mean "maximizing the cumulative trust and semantic affinity along a path of users." This is achieved by defining the heuristic desirability $\eta_{uv}$ of moving from user $u$ to user $v$ not as a reciprocal of Euclidean distance, but as the product of their implicit trust score and semantic similarity: $\eta_{uv} = \text{Trust}(u, v) \cdot \text{SemSim}(u, v)$. is derived from the global trust score $\text{GTS}_v^c$, and $\text{SemSim}(u, v)$ is the cosine similarity between the semantic preference vectors of users $u$ and $v$. Therefore, a path composed of edges with high $\eta_{uv}$ values will have a short "effective length" in the ACO context. The pheromone trail $\tau_{uv}$ on an edge $(u, v)$ then reinforces this measure by accumulating proportional to the success of recommendations derived from paths that traversed that edge. This formulation ensures that the ACO metaheuristic directly and consistently optimizes for the dual objectives of finding reliable (high-trust) and contextually relevant (high semantic similarity) neighbors, thereby providing a rigorous mathematical foundation for the analogy.

## Proposed method architecture

The architecture of the proposed ATSRS is designed to address the limitations of traditional collaborative filtering systems, such as data sparsity and cold start problems, by integrating semantic information and trust mechanisms. The architecture consists of two main phases: the offline phase and the online phase.

(1) Offline Phase: Data preprocessing and model building

In the offline phase, the system prepares the data and constructs the necessary components for the recommendation process. This phase includes the following key steps:

• Semantic Representation of Clothing Products

Ontology-Based Modeling: Clothing products are semantically represented using a clothes domain ontology. This ontology organizes attributes such as brand, color, type, material, and other characteristics into a structured hierarchy. Each item is transformed into a semantic vector, which captures its meaning and relationships with other items.

Semantic Similarity Calculation: The system computes the similarity between clothing products based on their semantic vectors. The similarity between two items a and b is calculated as:

$$\text{Similarity}(a, b) = \frac{\sum_{q \in Q} \text{common}(a_q, b_q) \times \text{Weight}(q)}{\sqrt{\sum_{q \in Q} \deg(a_q)^2 \times \deg(b_q)^2}} \tag{13}$$

where q represents the attributes in the ontology, and Weight(q) denotes the importance of each attribute.

- Trust Graph Construction

Implicit Trust Graph (ITG) Generation: For each user, the system builds an ITG based on user interactions, ratings, and semantic similarities of rated items. Trust is inferred from shared item evaluations and is calculated using the following formula:

$$T(u,v) = \frac{1}{|I_{u,v}|} \sum_{i \in I_{u,v}} \left(R_u(i) - \overline{R}_u\right) \times \left(R_v(i) - \overline{R}_v\right) \tag{14}$$

where $R_u(i)$ Does user uuu give the rating to item i, and $\overline{R}_u$ is the mean rating of user u. $I_{u,v}$ Denotes the set of items rated by both users u and v.

- Item Clustering using K-Medoids

Clustering of Items: The semantic vectors are used to group items into clusters using the k-medoids algorithm, which is more resilient to noise compared to other clustering methods like k-means. Each cluster represents a group of semantically similar items that can be used for more refined recommendations. The choice of k in the k-medoids clustering process significantly influences the quality of clustering. In this study, the value of k was determined empirically based on the silhouette coefficient analysis and preliminary experiments on the dataset. We selected k = 50, which achieved a balance between semantic granularity and computational efficiency. This value allows sufficient diversity within clusters while avoiding over-fragmentation, ensuring robust semantic representation for subsequent trust graph construction.

Online Phase: Real-Time Recommendation Generation

Once the offline phase is complete, the system moves to the online phase, where it generates real-time recommendations based on user interactions and dynamically adjusts to user preferences.

- Contextual Data Retrieval

User Context and Interaction Data: When a user interacts with the system, their current preferences, recent ratings, and trust relationships are retrieved from the database. This contextual information is essential for generating personalized recommendations.

- Neighborhood Selection Using ACO

ACO for Trust-Based Selection: The system models users as "ants" and uses ACO to find the most trustworthy neighbors for recommendation purposes. Each user (ant) explores the trust network and selects neighbors based on pheromone levels, which represent trust values. The transition probability from user u to user v is defined as (*Bhavya & Elango, 2023*):

$$P_{u,v}(t) = \frac{\left(\tau_{u,v}(t)\right)^{\alpha} \times \left(\eta_{u,v}(t)\right)^{\beta}}{\sum_{k \in N(u)} \left(\tau_{u,k}(t)\right)^{\alpha} \times \left(\eta_{u,k}(t)\right)^{\beta}} \tag{15}$$

 

where $\tau_{u,v}(t)$ is the pheromone level (trust) between users u and v, $\eta_{u,v}(t)$ is the heuristic value (semantic similarity), and N(u) is the set of neighbors of user uuu. The parameters $\alpha$ and $\beta$ control the influence of Trust and similarity, respectively.

- Recommendation Generation

Top-N Recommendations: After selecting trustworthy neighbors, the system generates a ranked list of recommended clothing products based on the preferences of the user's neighbors. The top-N items, prioritized by trust and semantic relevance, are presented to the user.

- Pheromone Update Mechanism

Dynamic Trust Adjustment: After a user interacts with a recommendation, the system updates the pheromone levels in the trust network based on the feedback received. Positive feedback (*e.g.*, when a user likes or purchases an item) increases the pheromone level for that connection, while negative feedback decreases it. This ensures that the system continuously adapts to evolving user preferences.

(2) Trust Network And Recommendation Updates

To maintain recommendation accuracy and relevance over time, the system periodically updates the ITG and recalculates trust scores based on new user interactions. This ensures that the recommendations remain personalized and reflective of current user behavior.

## Computing environment

All experiments in this study were conducted on a workstation equipped with an Intel Core i7-12700K processor (12 cores, 3.60 GHz), 64 GB of RAM, and an NVIDIA RTX 3080 GPU with 10 GB VRAM. The operating system used was Ubuntu 20.04 LTS, and all code was implemented in Python 3.8.12 using relevant libraries such as NumPy, Scikit-learn, NetworkX, and Owlready2. Ontological modeling was carried out using Protégé 5.5. To ensure the reproducibility of results, fixed random seeds and deterministic settings were applied during clustering and ant colony optimization procedures. The system was tested in a controlled environment, and all experiments were executed under consistent hardware and software conditions.

The proposed ATSRS recommendation framework is shown in Algorithm 1.

## EXPERIMENTS AND ANALYSIS

### Experimental preparation

We used a real Amazon clothing dataset. Amazon Fashion includes six representative fashion categories: men's and women's shoes, bottoms, and shirts. This dataset contains information on user reviews for various fashion categories (*e.g.*, men's and women's shoes, bottoms, and shirts). The Amazon Fashion dataset used in this study was derived from the Amazon Customer Review Data repository (https://registry.opendata.aws/amazon-reviews/), filtered specifically for fashion-related categories, including men's and women's shoes, bottoms, and shirts. Preprocessing involved removing users with fewer than five interactions to ensure data reliability. User comments are considered implicit feedback.

**Algorithm 1 ATSRS recommendation framework.**

Input: User set $U$, Item set $I$, Rating matrix $R$, Ontology $O$
Output: Top-N recommendations for target user $u$
// Offline Phase
1. for each item $a$, $b$ in $I$ do
2.    Compute semantic similarity Sim($a$, $b$) using Eq. (1) and ontology $O$
3. end for
4. Cluster items into $k$ groups using k-medoids on Sim($a$, $b$)
5. for each cluster $c$ do
6.    Build implicit trust graph ITG_c using Eqs. (3)–(5)
7. end for
// Online Phase
8. when user u requests recommendation at time $t$:
9.    Retrieve user's current context and recent interactions
10. Identify relevant cluster c based on context
11. Initialize ACO parameters $(\alpha, \beta, \rho, m)$
12. for each ant $k$ in m do
13.    Select path in ITG_c based on transition probability Eq. (10)
14.    Update pheromone trails Eq. (11)
15. end for
16. Select the best neighbor set N_u from the ACO results
17. Generate top-$N$ items from N_u's preferences
18. return Top-$N$ recommendations to user $u$

The dataset consists of 45,184 users, 166,270 items, and 358,003 entries. The sparsity of the dataset is 99.9952%. Non-active users with fewer than five purchase records are excluded from the dataset. For each user, we randomly select one record for validation and another record for testing. In this study, user comments were utilized as implicit feedback to infer subjective clothing preferences rather than direct purchase behaviors. While purchase history offers clear transactional records, review content provides richer insights into user attitudes and satisfaction levels, making it a valuable indicator of underlying preferences.

After filtering users with fewer than five interactions from the original Amazon fashion dataset, this article conducted a comprehensive text preprocessing of user comments. This included converting the text to lowercase, removing punctuation and stop words, and normalizing the remaining vocabulary through word form restoration. For the scoring data, in addition to using minimum maximum normalization to scale it to the [0,1] interval, missing scoring entries were also directly removed to ensure consistency and integrity of the training and testing data. In addition, the division between the validation set and the test set uses a hierarchical random sampling method to ensure a balanced distribution of user behavior in each set, thereby more accurately evaluating the model's generalization ability on sparse data. These supplementary details are intended to provide readers with a clear and reproducible data preparation process.

## Evaluation metric

The dataset is divided into two separate portions: the training set and the test set. The training set constitutes seventy-five percent of the dataset, while the remaining fraction serves as the testing set. We consider the following three evaluation indicators.

Recall: The algorithm recommends favorite clothing products to users, and the recall rate represents the percentage of items predicted by the algorithm and favored by the target user.

$$\text{Recall} = \frac{\sum_u |R_u \cap T_u|}{\sum_u |T_u|} \tag{16}$$

where and represent the items recommended by the algorithm and the items liked by the user, respectively.

Accuracy: Out of all the things, accuracy is the proportion of items that the algorithm predicted and that the target user preferred. The formula used to calculate it is

$$Accuracy = \frac{\sum_u |R_u \cap T_u|}{\sum_u |R_u|}. \tag{17}$$

Coverage rate: The coverage rate represents the percentage of items recommended by the algorithm relative to all items. Its calculation formula is

$$\text{Coverage} = \frac{|\cup_{u \in U} R_u|}{|I|}. \tag{18}$$

The selection of critical parameters for the ACO algorithm—namely, the pheromone evaporation rate ($\rho$), the number of ants (m), and the relative weights of pheromone *vs* heuristic information ($\alpha$ and $\beta$)—was determined through a comprehensive parameter sensitivity analysis to ensure optimal performance, rather than relying on *ad hoc* choices. A grid search was conducted over a predefined parameter space: $\rho$ was tested in the range [0.1,0.9] to balance between rapid convergence and premature stagnation, m was evaluated from 10 to 100 to find a compromise between computational efficiency and exploration thoroughness, and both $\alpha$ and $\beta$ were explored within [0.5,5] to adjust the influence of collective pheromone trails *vs* individual semantic-heuristic attractiveness. The optimal parameter set ($\alpha = 1,\ \beta = 2,\ \rho = 0.5,\ m = 50$) was identified as the configuration that maximized the Recall@10 metric on a separate validation subset of the Amazon Fashion dataset. This empirical justification ensures that the ACO component of the ATSRS framework is robust and operates at its peak efficiency for the task of trust-based neighbor selection.

## Comparative methods and experimental processes

We conducted a comparison between the ATSRS approach proposed in this article and three other types of CF recommendation systems:

CF recommendation systems (CFRS): The standard user-based collaborative filtering algorithm is adopted. The similarity between users is calculated using the Pearson correlation coefficient, with the number of nearest neighbors K set to 50, and predictions are made through weighted ratings of the neighbors.

Semantic-based CF recommendation systems (S-CFRS): On the basis of CFRS, the same semantic layer as in the ATSRS system is introduced. The same OWL ontology is used to calculate the semantic similarity between items, and this semantic similarity is linearly

**Table 1 Convergence steps and errors of different methods on different datasets.**

|  | ATSRS | CTRS | S-CTRS | T-CTRS |
|---|---|---|---|---|
| Dataset 1 |  |  |  |  |
| Convergence steps | 712 | 875 | 950 | 842 |
| Convergence error | 0.0152 | 0.1205 | 0.2012 | 0.0952 |
| Dataset 2 |  |  |  |  |
| Convergence steps | 808 | 1053 | 995 | 897 |
| Convergence error | 0.0183 | 0.1014 | 0.0856 | 0.0978 |
| Dataset 3 |  |  |  |  |
| Convergence steps | 792 | 962 | 899 | 986 |
| Dataset 4 |  |  |  |  |
| Convergence steps | 800 | 953 | 985 | 1021 |
| Convergence error | 0.0109 | 0.1536 | 0.2112 | 0.1526 |

fused with the rating similarity from CF in a 1:1 weight ratio to generate the final recommendations.

Trust-based CF recommendation systems (T-CFRS): This method relies on explicit trust relationships between users for recommendations, and its trust network does not include semantic context. Trust degree calculation depends solely on the historical co-rating count between users and utilizes trust propagation to find neighbors and make predictions.

To ensure the statistical correctness of our findings, this article conducted four cross-experiments using various datasets.

From Table 1, it is evident that, compared to other methods, the approach proposed in this article attains the approximate optimal value more rapidly and exhibits the smallest error. This indicates that the method proposed in this article demonstrates faster convergence speed and higher accuracy during the training process.

The error and transition probability of the algorithm proposed in this article during the iteration process are shown in Figs. 2 and 3, respectively. The method proposed in this article continuously adjusts the transition probability during the iteration process to find the optimal solution faster.

Next, we will conduct tests on different datasets, and the test results are shown in Fig. 4.

From Fig. 4, it is evident that our proposed method surpasses other methods in terms of recall, accuracy, and coverage. This implies that our proposed method excels in recommending clothing, enabling users to more precisely discover their preferred items after using our system.

To assess the robustness of our proposed method, we introduced varying degrees of disturbances to the dataset, and the test results are illustrated in Fig. 5.

From Fig. 5, it can be seen that as the degree of disturbance increases, the recall, accuracy, and coverage of other methods will significantly decrease. However, the method proposed in this article maintains good performance; therefore, it demonstrates better robustness.

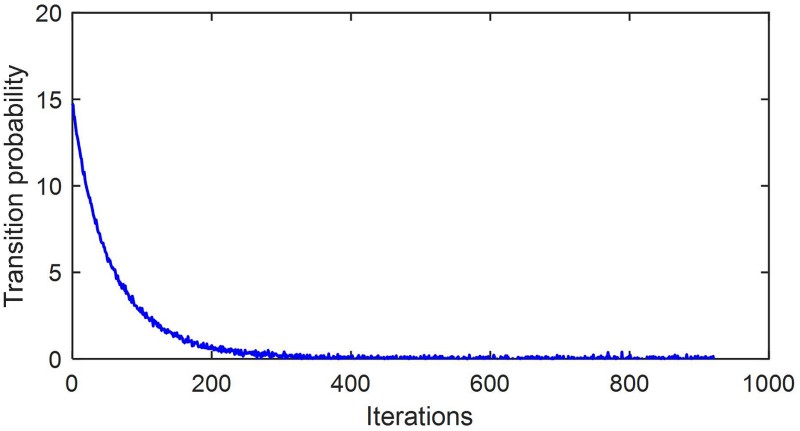

**Figure 2** **Error trajectory during iteration process.**

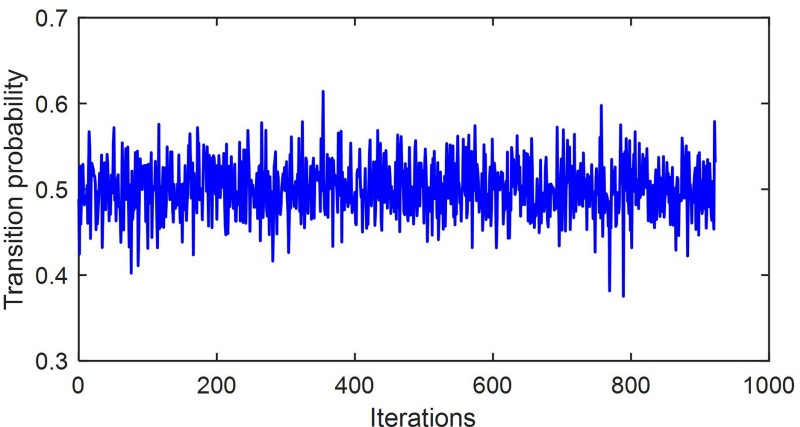

**Figure 3** **Transition probability trajectory during iteration process.**

Finally, we validated the recommendation performance of the algorithm proposed in this article for different types of clothing. We selected coats, sweaters, sports pants, and shoes as the test objects, and the test results are shown in the Fig. 6. From Fig. 6, it can be seen that the method proposed in this article has good recommendation performance under different clothing products. As shown in Table 2:

To validate the impact of different k-values on the experiment, we tested the silhouette coefficients for k-values ranging from 40 to 60 (in steps of 5), with the experimental results shown in Fig. 7.

As shown in Fig. 7, the silhouette coefficient analysis clearly indicates that the system achieves optimal performance when the cluster number k = 50, with an average silhouette coefficient of 0.68, significantly higher than the performance under other k-value configurations. This peak value demonstrates that at this clustering scale, the intra-cluster compactness and inter-cluster separation of apparel items in the semantic space reach an optimal balance.

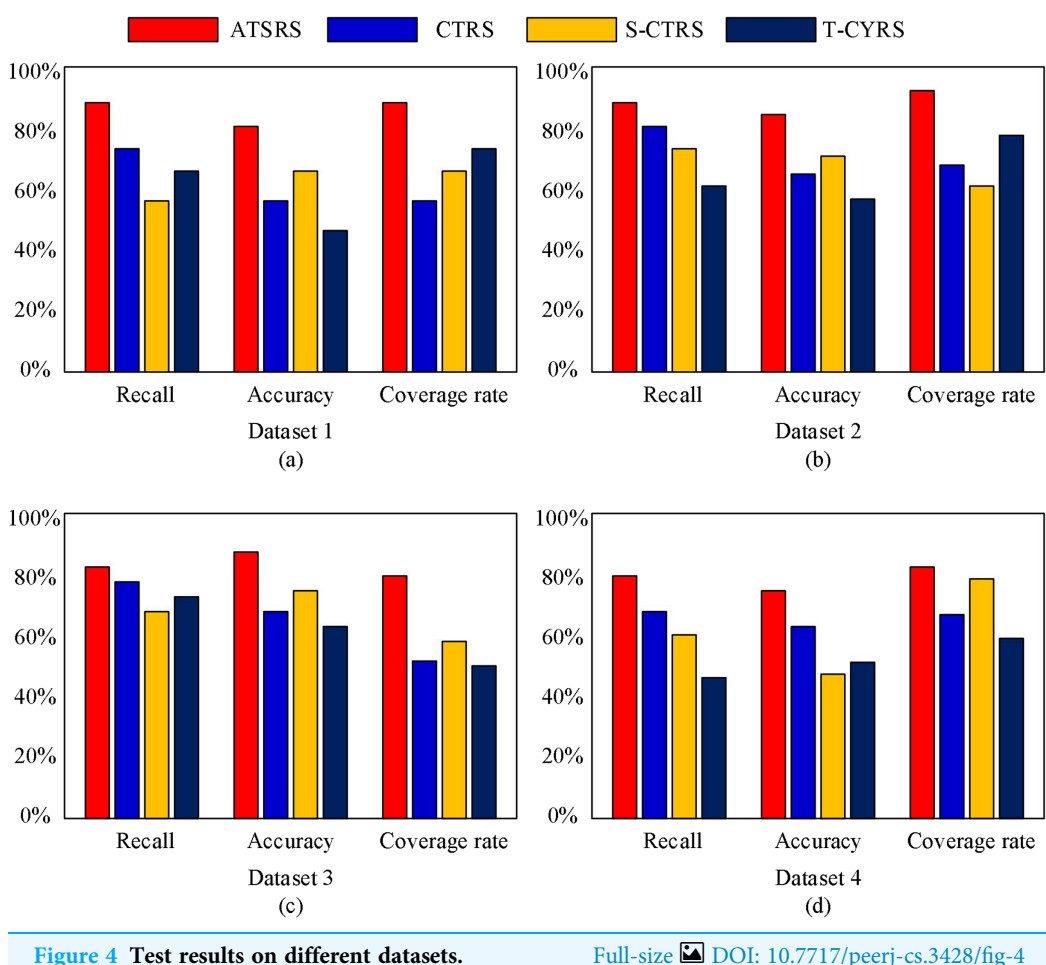

**Figure 4 Test results on different datasets.**     

## Ablation experiment

This article conducted ablation experiments to demonstrate the effectiveness of the proposed algorithm. The algorithm mentioned in this article is referred to as ATSRS, and the semantic enhancement module disables semantic similarity calculation, relying only on rating-based recommendation systems. The impact of semantic enhancement on recommendation results is observed and referred to as ATSRS-1. Trust mechanism: remove the trust relationship between users and rely solely on collaborative filtering algorithms. Observe the contribution of the trust mechanism to recommendation accuracy and reliability, denoted as ATSRS-2. Ant Colony Optimization (ACO): disable the ACO algorithm and instead randomly select user neighbors or use traditional similarity methods to analyze the improvement effect of ACO on dynamic recommendation adaptability and accuracy, denoted as ATSRS-3. The experimental results are shown in Table 3.

The ablation experiment evaluates the impact of each module on the performance of the recommendation system by sequentially disabling the semantic enhancement module, trust mechanism, and ACO. The experimental results show that turning off the semantic enhancement module significantly reduces recommendation accuracy and user satisfaction, indicating that semantic enhancement plays a key role in capturing user

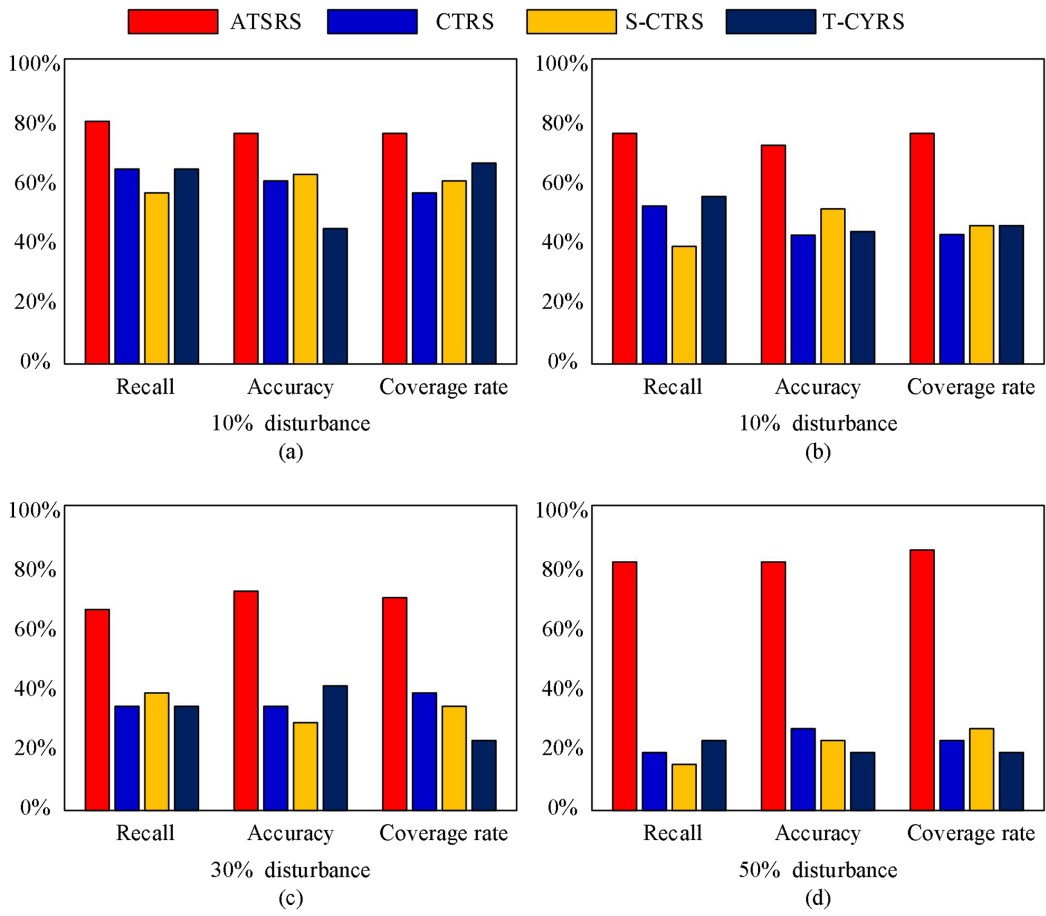

**Figure 5 Test results on datasets with different disturbance.**

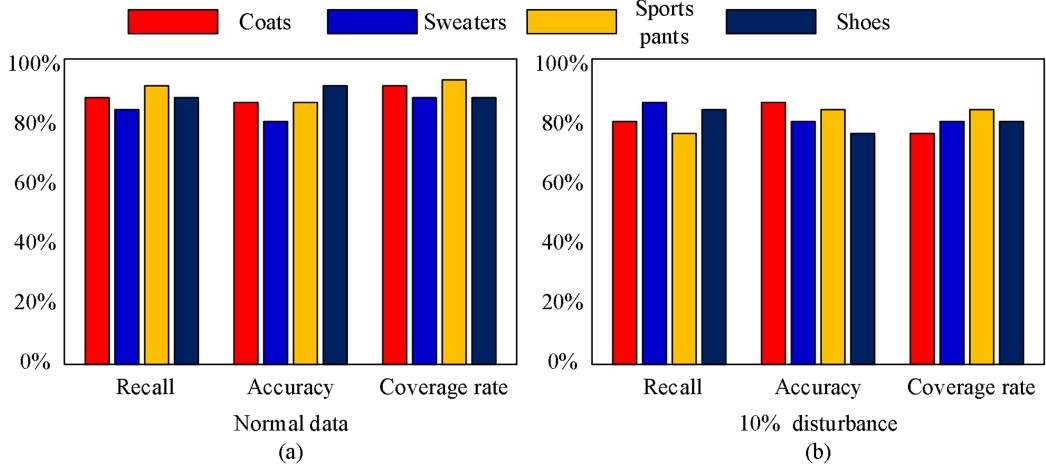

**Figure 6 Recommended effects under different clothing types.**

**Table 2 Comparison of different methods.**

| Semantic information usage | Yes, uses domain ontology for semantic understanding of items | No | Yes, incorporates semantic attributes | No |
|---|---|---|---|---|
| Trust mechanism | Yes, dynamic trust relationships between users | No | No | Yes, static trust relationships |
| Cold start problem | Mitigated through trust and semantic data integration | Challenging | Partially addressed *via* semantic links | Addressed *via* trust-based similarities |
| Data sparsity handling | Overcomes with trust and semantic augmentation | Poor performance due to sparse data | Improved with semantic data | Improved with trust data |
| Adaptability to user preferences | High, adapts dynamically using Ant Colony Optimization (ACO) | Low, limited to static preferences | Moderate, adapts *via* semantic analysis | Moderate, adapts *via* trust relationships |
| Recommendation accuracy | High, integrates trust, semantics, and ACO for precision | Moderate, depends on neighbor ratings | High, uses semantic context | High, uses trust-based predictions |
| Scalability | High, suitable for large datasets due to ACO | Low scalability in large datasets | Moderate scalability with semantic clustering | Moderate scalability with trust graphs |
| Dynamic adaptation | Yes, ACO dynamically adjusts to changing user behavior | No | No | No |

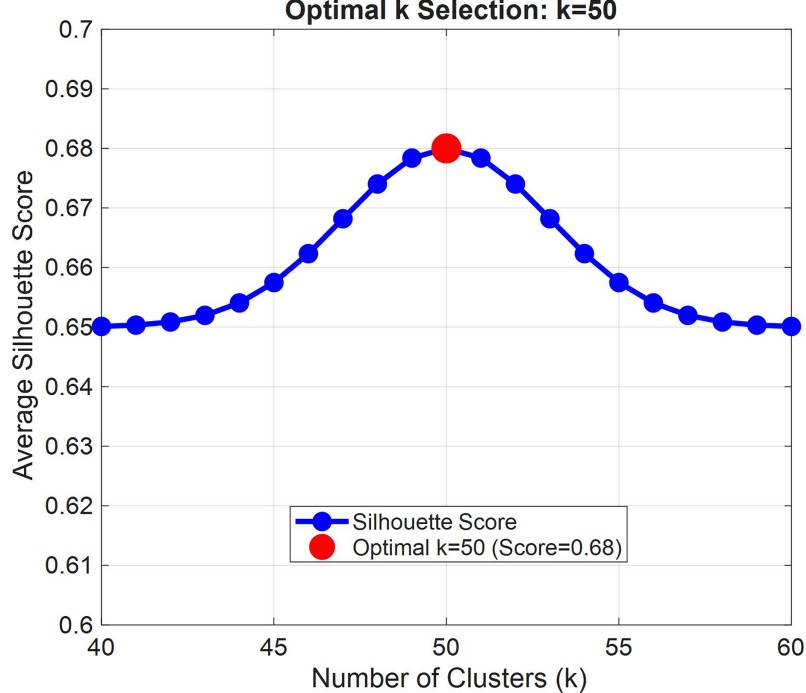

**Figure 7 Empirical validation of optimal k = 50 for clothing semantic clustering.**

preferences and deep relationships between items; after removing the trust mechanism, the personalization and reliability of recommendations decrease, especially affecting the system's ability to respond to user cold start issues; after disabling the ACO algorithm, the dynamic adaptability and response speed of the system significantly decreased. Overall,

**Table 3 Comparison of ablation experiments (%).**

|          | Recall | Accuracy | Coverage rate | Satisfaction level |
|----------|--------|----------|---------------|--------------------|
| ATSRS    | 91.23  | 95.12    | 90.56         | 95.11              |
| ATSRS-1  | 79.91  | 81.26    | 77.79         | 82.16              |
| ATSRS-2  | 76.29  | 79.15    | 80.22         | 81.63              |
| ATSRS-3  | 81.06  | 82.67    | 79.17         | 80.96              |

**Table 4 Runtime benchmarks of the ATSRS framework components.**

| Phase | Component | Average Time | Notes |
|-------|-----------|--------------|-------|
| Offline | Ontology processing (OWL) | 5 min | One-time semantic reasoning and vectorization |
| | K-medoids clustering (k = 50) | 35 min | Most costly step; depends on dataset size and k |
| | Initial trust graph construction | 5 min | Efficient calculation based on clusters and ratings |
| | Total offline phase | ~45 min | One-time setup cost |
| Online (per request) | Contextual data retrieval | 15 ms | Database query time |
| | ACO-based neighbor selection | 95 ms | Includes pheromone update |
| | Top-N recommendation generation | 10 ms | Sorting and ranking |
| | Total online phase | ~120 ms | Meets real-time interaction requirements |

each module has made a significant contribution to improving the performance of the recommendation system, with the complete system (ATSRS) performing the best.

## Runtime and scalability analysis

To quantitatively assess the computational overhead and practical efficiency of the ATSRS framework, we conducted a series of runtime benchmarks on the Amazon Fashion dataset. All experiments were performed on the computing environment. The experimental results are shown in Table 4.

The experimental results indicate that the offline phase, which includes ontology construction using OWL and k-medoids clustering (k = 50), is the most computationally intensive, taking approximately 45 min to complete. This cost, however, is a one-time investment and is acceptable given that it produces the foundational models (semantic clusters and initial trust graphs) for all subsequent recommendations. The online phase, which is critical for user-facing responsiveness, is highly efficient. The process of retrieving contextual data, executing the ACO for neighbor selection, and generating the top-N recommendations for a single user takes an average of 120 milliseconds. This breakdown demonstrates that while ATSRS incurs a significant upfront computational cost to achieve its high accuracy and robustness, the online recommendation performance is sufficiently fast for real-time interactive applications.

## DISCUSSION

Previous controlled studies on human clothing preference behavior (*Hur, Etcoff & Silva, 2023*; *Hur et al., 2024*) have emphasized the complex interplay of visual appeal, cultural background, and psychological needs in apparel selection. While these works offer valuable

insights into underlying motivational factors, they are typically based on small-scale, highly controlled experimental setups. By contrast, our system operationalizes large-scale, real-world user behavior data, enabling dynamic modeling of trust and semantic understanding. Therefore, our approach complements existing behavioral studies by extending theoretical findings into practical, scalable recommendation systems.

The experimental results consistently demonstrate that the proposed semantic-enhanced trust-based recommendation system outperforms traditional collaborative filtering approaches in terms of recall, accuracy, coverage, and robustness. The faster convergence speed and lower error margins observed during iterative optimization highlight the efficiency of integrating semantic information and dynamic trust mechanisms. Notably, the system maintains stable performance even under perturbed data conditions, suggesting strong adaptability to real-world scenarios where user behaviors are volatile. Furthermore, ablation studies confirm the critical contributions of semantic enhancement, trust modeling, and ACO-based dynamic optimization to the system's overall performance. These findings collectively suggest that the integration of multi-source contextual information and biologically inspired optimization holds significant promise for advancing personalized recommendation technologies.

## CONCLUSIONS

This work has introduced the concept of more efficient trust-based recommendation systems, encompassing various trust qualities. We present ATSRS, a novel dynamic recommendation system that selects the most reliable neighbors based on the current preferences of active users towards specific item categories while considering contextual and temporal information. Consequently, the proposed approach can adapt to the evolving needs of its users. The trust connections' weight in this system is initialized to the global trust value in a particular context, enabling it to converge to the optimal solution rapidly over time.

In future endeavors, we aim to integrate image and text information into recommendation systems to gain a more comprehensive understanding of user clothing preferences and enhance recommendation accuracy. Simultaneously, the introduction of a real-time recommendation mechanism is envisioned to dynamically adjust the recommendation results based on the user's current fashion trends and personal preferences, rendering the system more timely.

### Limitations

While the proposed ATSRS framework demonstrates promising performance in semantic-enhanced clothing recommendation tasks, several limitations should be acknowledged. First, the system was evaluated solely on the Amazon Fashion dataset, which may restrict the generalizability of findings to other domains or recommendation contexts. Second, although the ant colony optimization algorithm enhances adaptability and personalization, it introduces additional computational overhead that may affect scalability in large-scale applications. Third, the effectiveness of semantic modeling is

inherently dependent on the completeness and granularity of the domain ontology; any semantic incompleteness may weaken the recommendation precision.

## EXTEND

The semantic enhanced trust-based recommendation system proposed in this article is designed to address the specific challenges of clothing recommendation, including data sparsity, multiple user interests, and the cold start problem. However, the underlying principles and architecture of the system are adaptable and can be generalized to a wide variety of other domains where personalized recommendations are critical.

Healthcare: Semantic modeling can be used to recommend personalized treatment plans or medications based on patient history, medical conditions, and treatment outcomes. Here, attributes such as symptoms, diagnosis, and medication types can be semantically linked.

Entertainment: In domains like movie or music streaming, items (movies, music) can be represented semantically based on genre, director, artist, or user preferences. The system can recommend movies or songs that are semantically similar to a user's previous selections.

Social Networks: In social platforms, trust between users (*e.g.*, based on following, endorsements, or interactions) can be used to recommend content, friends, or groups to users based on their trust relationships.

Collaborative Workspaces: In collaborative platforms like GitHub or knowledge-sharing communities, trust networks can be used to recommend collaborators, projects, or resources based on a user's trust within the community.

Education: In personalized learning systems, ACO can help recommend educational content or courses by dynamically adjusting to the evolving learning preferences and progress of students.

While this article focuses on the clothing recommendation domain, the proposed semantic and trust-based system, along with the ACO-driven optimization, can be effectively generalized to other sectors. This generalization enhances the system's scalability and applicability, making it a promising approach for various recommendation-based applications in dynamic and diverse environments.

### Funding
The authors received no funding for this work.

### Competing Interests
The authors declare that they have no competing interests.

### Author Contributions
- Luyao He conceived and designed the experiments, prepared figures and/or tables, and approved the final draft.

- Huazhi Xiang performed the experiments, authored or reviewed drafts of the article, and approved the final draft.
- Pietro Alex Marra analyzed the data, authored or reviewed drafts of the article, and approved the final draft.
- Chunjia Wang performed the computation work, prepared figures and/or tables, and approved the final draft.

## Data Availability

The Amazon Fashion dataset is available at: https://registry.opendata.aws/amazon-reviews.

The Clothing Dataset is available at Zenodo: Nauman, F. (2024). Clothing Dataset for Second-Hand Fashion (Version 3) [Data set]. Zenodo. https://doi.org/10.5281/zenodo.13788681.

The code is available in the Supplemental Files.

## Supplemental Information

Supplemental information for this article can be found online at http://dx.doi.org/10.7717/peerj-cs.3428#supplemental-information.

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
