# Peer review of "A semantic enhanced clothing recommendation system based on implicit trust graphs and ontology web language"

_PeerJ Computer Science, doi:10.7717/peerj-cs.3428_

## Round 0.1 · original submission · Major Revisions

· Academic Editor

Major Revisions

Thank you for submitting your manuscript. We appreciate the effort and innovation demonstrated in your research, which addresses critical challenges in online clothing recommendation systems, including data sparsity, cold start issues, and the need for personalized suggestions across diverse categories.

After a thorough peer review process from experts, I would like to inform you that your manuscript needs a Major revision as outlined in the reviewer comments.

Academic Editor Comments:

Please give a clearer explanation of the ant colony optimization parameters (e.g., pheromone updates, evaporation rates) to enhance reproducibility.

A detailed comparison with traditional methods (e.g., collaborative filtering) using quantitative metrics like precision and recall would strengthen the claims of the paper.

Please improve the language of the manuscript while resubmitting.

**Language Note:** The review process has identified that the English language must be improved. PeerJ can provide language editing services - please contact us at [email protected] for pricing (be sure to provide your manuscript number and title). Alternatively, you should make your own arrangements to improve the language quality and provide details in your response letter. – PeerJ Staff

Reviewer 1 ·

Basic reporting

The manuscript presents a semantic-enhanced clothing recommendation framework (ATSRS) that integrates ontology-based semantic modeling, implicit trust graphs, and an ant colony optimization mechanism. The overall architecture is interesting and combines well-established concepts from recommender systems and swarm intelligence. However, the technical rigor and reproducibility of the proposed framework remain insufficiently validated.

Experimental design

The description of the ATSRS architecture is rather high-level and descriptive. While Figure 1 outlines the offline and online phases, the workflow lacks pseudocode or a formal algorithm description.
 The semantic similarity calculation formula is introduced but remains underspecified: terms such as deg(a,q) and common(a,b,q) are mentioned without formal mathematical definition or constraints. A detailed explanation of how weights Weight(q) are determined (manual assignment vs. learning-based) is also missing.
 The process of constructing implicit trust graphs is presented, yet the derivation of local and global trust scores (Eq. 3 and subsequent formulas) lacks theoretical validation.
 Although the ant colony optimization is claimed to optimize neighborhood selection, critical parameters such as pheromone evaporation rate, number of ants, and α/β weights are not justified.
 In Section 3.4, Ant Colony Optimization is employed to select the “best neighbor,” yet the analogy between path length in ACO and trust distance between users is not formally justified. Minimizing path length does not necessarily equate to identifying the most reliable or informative neighbor. The mapping between pheromone trails, heuristic similarity, and trust weights requires a clearer mathematical formulation to avoid conceptual inconsistency.
 In Section 3.2, when users have no co-rated items, the system simply assumes “no trust relationship,” which exacerbates sparsity, particularly under cold-start conditions. This design choice risks leaving many users as isolated nodes in the trust graph. Incorporating indirect trust propagation or semantic similarity-based inference could reduce sparsity and improve the robustness of the trust network.
 The pheromone update rule is presented but not adapted for the recommendation setting. For example, the mapping between “path length” in ACO and “user similarity/trust” is only loosely defined, which weakens the theoretical consistency of the adaptation.
 Ontology construction with OWL and k-medoids clustering (Section 3.2) may introduce high computational overhead. The manuscript lacks runtime benchmarks,

Validity of the findings

the technical rigor and reproducibility of the proposed framework remain insufficiently validated.

·

Basic reporting

The manuscript is well structured, but the language is often unclear, ambiguous, and does not meet professional standards. There are numerous typographical, awkward phrasing, and grammatical issues throughout, which make comprehension difficult for an international audience. For instance, the title refers to a "clothing design system," but the abstract and body consistently describe a recommendation system; this inconsistency needs clarification. I recommend a thorough review by a proficient English speaker or professional editing service, focusing specifically on lines 16-38 in the abstract and sections 1-3. The introduction provides adequate context on online clothing shopping challenges, such as data sparsity and cold start problems, and references relevant literature on collaborative filtering and hybrid systems. The motivation is clear: addressing limitations in traditional recommendation systems through semantic enhancement, trust mechanisms, and ant colony optimization. The literature is reasonably referenced, although some citations are duplicated (e.g., references 29 and 30), and more recent works on semantic embeddings could strengthen the background. The structure generally conforms to PeerJ standards for an AI application article, with sections on introduction, related works, methodology, experiments, discussion, and conclusions. Deviations, such as the "Extend" section (possibly meant as "Extensions"), do not significantly improve clarity and could be integrated into the discussion or future work. Formal results include definitions of terms like semantic similarity and trust scores, with formulas provided, but detailed proofs are absent where necessary (e.g., for the convergence of the ACO algorithm).

Experimental design

The article is within the aims and scope of PeerJ Computer Science as an AI Application, focusing on a recommendation system using semantic, trust, and optimization techniques. The investigation appears conducted to a reasonable technical standard, with ethical considerations implicitly met (e.g., using public datasets without personal data issues).
Methods are described with moderate detail, including semantic modeling via OWL ontology, trust graph construction, K-Medoids clustering (k=50 selected empirically), and ACO for neighbor selection. Replication is partially feasible: the Amazon Fashion dataset is public (link provided), computing environment is specified (Python 3.8 with libraries like NumPy, Scikit-learn), but no code, reproduction script, or full ontology is shared, limiting full reproducibility.
Data preprocessing is discussed briefly: filtering users with fewer than 5 interactions, random selection for validation/testing, and treating reviews as implicit feedback. This is sufficient for the context but could be expanded (e.g., handling missing values or normalization specifics beyond min-max scaling).
Evaluation methods (recall, precision, coverage), assessment metrics, and model selection (e.g., silhouette coefficient for k) are adequately described with formulas. Comparisons are made to baselines like CFRS, S-CFRS, T-CFRS, with 4-fold cross-validation for statistical validity. Sources are cited appropriately, with paraphrasing where used.

Validity of the findings

Replication is encouraged but not assessed for novelty; the rationale for combining semantics, trust, and ACO is clearly stated, benefiting the field by addressing MIMC issues in clothing recommendations.
Conclusions are well stated and limited to supporting results, e.g., ATSRS shows faster convergence, higher accuracy (e.g., lower error in Table 1), and better robustness (Figures 4-5). Experiments are performed satisfactorily, including comparisons, ablation studies (Table 3), and tests under disturbances/clothing types (Figures 5-6). The argument is developed and meets introduction goals: improving personalization and reliability via collective intelligence.
The conclusion identifies limitations (e.g., dataset specificity, computational overhead) and future directions (e.g., integrating image/text, real-time mechanisms), which are appropriate.

Additional comments

This manuscript presents an interesting hybrid recommendation system for clothing, combining semantic ontology, implicit trust graphs, and ant colony optimization. Strengths include the innovative use of ACO for dynamic trust path selection, empirical validation on a real dataset, and ablation experiments demonstrating each component's value. The results are compelling, showing superior recall, precision, and coverage over baselines, with good robustness.
However, the primary weakness is the poor English quality, which hinders readability—e.g., repetitive phrasing in sections 3-4 and inconsistencies like "design" vs. "recommendation" in the title/abstract. The methodology could benefit from more depth on ontology construction and hyperparameter tuning. Figures (e.g., 2-6) appear clear but should be checked for manipulation; no issues evident from descriptions. References have duplicates and could include more on recent OWL applications.
I suggest major revisions: 1) Professional language editing throughout. 2) Clarify title to match content (e.g., "recommendation" instead of "design"). 3) Provide code repository for reproducibility. 4) Expand preprocessing discussion and add statistical tests (e.g., p-values) for results. 5) Remove reference duplicates.

---

## Round 0.2 · Minor Revisions

· Academic Editor

Minor Revisions

Dear author

Thank you for your revision submission.

Although majority of the comments are addressed and paper is improved well, however, there are couple of minor points to be addressed so that the paper should be further improved.

The comments are with this trailing email.

It is not necessary to cite the suggested literature if it's not relevant to your domain, please revise and resubmit for further evaluation.

Reviewer 1 ·

Basic reporting

The manuscript is well-structured, and the flow of ideas is logically organized from introduction to conclusion.

Experimental design

The experimental setup is well-described, allowing the study to be understood and replicated.

Validity of the findings

The discussion of results is logical and transparent, with interpretations that align well with the data presented.

Additional comments

no comments

·

Basic reporting

The manuscript is generally written in clear and understandable English, though there are occasional minor grammatical errors and awkward phrasings (e.g., "clothing projects" is used frequently where "clothing items" or "apparel products" would be more standard). The structure largely conforms to disciplinary norms.

However, a significant issue affects basic reporting: several key figures and tables are missing or incomplete in the provided PDF. Specifically:

Figure 1 is present and clear.

Figures 2, 3, 4, 5, and 6 are referenced in the text but their content is not fully visible or is missing from the PDF pages (e.g., Page 30-34 show only axes and legends without data lines or bars).

Table 1 is partially complete but missing data for Datasets 3 and 4 in the "Convergence error" rows.

Tables 2, 3, and 4 are referenced as "on next page" but are not present in the subsequent pages of the uploaded file.

Suggested Improvements:

The authors must provide a complete manuscript with all figures and tables fully rendered and legible. The data in the figures is critical for evaluating the experimental results and claims of superiority.

A thorough proofreading by a native or highly proficient English speaker is recommended to polish minor language issues.

Experimental design

The experimental design is sound in concept. The article falls within the journal's scope, the methodology is described in substantial detail, and the use of a real-world dataset (Amazon Fashion) is appropriate. The description of data preprocessing (Section 4.1) is sufficient, and the evaluation metrics (Recall, Accuracy, Coverage) are standard and well-defined. The ablation study is a particular strength, effectively isolating the contribution of each proposed module.

Areas for Improvement:

Reproducibility: While the computing environment is described, the manuscript would be significantly strengthened by including a link to the source code and the specific, preprocessed version of the dataset used. The current Data Availability link points to Zenodo, but it is unclear if it contains code or just data.

Parameter Justification: The choice of k=50 for k-medoids clustering is mentioned as being based on the silhouette coefficient, but no results of this analysis are shown. Providing a brief summary or a reference to a supplementary figure would bolster this claim.

Comparative Methods: The description of the baseline methods (CFRS, S-CFRS, T-CFRS) is too brief. For replicability and fairness, the authors should more precisely define the implementation details of these baselines (e.g., which specific similarity measure was used for CFRS, which semantic model for S-CFRS).

Validity of the findings

The conclusions about the performance of ATSRS are promising but cannot be fully validated with the current submission due to the missing figures. The text states that ATSRS outperforms baselines in recall, accuracy, coverage, and robustness, and the ablation study shows a clear drop in performance when key components are removed. This internal consistency supports the authors' argument.

Areas for Improvement:

The core claim of outperforming baselines rests on the data in Figures 4 and 5 and Tables 1-3. These must be provided and clearly labeled for the findings to be considered valid.

The argument that ACO is superior to other bio-inspired algorithms is supported by a citation, but a more detailed justification specific to the recommendation task would strengthen the manuscript.

The limitations section is well-stated and appropriate.

The conclusions are supported by the results described in the text and do a good job of identifying future directions.

Additional comments

This manuscript presents a novel and well-motivated approach by integrating semantic ontology, dynamic trust graphs, and ant colony optimization. The architecture is thoughtfully designed with separate offline and online phases, making it practical for real-world deployment. The ablation study is a key strength that convincingly demonstrates the value of each component. Once the missing experimental results (figures and tables) are provided and the methodological descriptions are fleshed out, this could be a strong contribution to the field of recommender systems.

To further strengthen the literature review in Section 2 (Related Works), I recommend citing the following relevant work that addresses cold-start problems and compares collaborative filtering approaches:

Ahmed, E., & Letta, A. (2023). Book Recommendation Using Collaborative Filtering Algorithm. Journal of Engineering, 2023, 1514801. https://doi.org/10.1155/2023/1514801

This study provides a comparative analysis of memory-based and model-based collaborative filtering, specifically demonstrating that matrix factorization techniques (SVD) outperform KNN-based approaches in handling cold-start scenarios. Given that your work also addresses cold-start problems through semantic and trust mechanisms, this reference would provide valuable context and support for your methodological choices.

---

## Round 0.3 · accepted · Accept

· Academic Editor

Accept

Thank you for your revision, I'm pleased to notify that your manuscript is accepted for publication.

Thank you for your fine contribution